# Structural maintenance of chromosomes (SMC) proteins are required for DNA elimination in *Paramecium*

Fukai Zhang, Sebastian Bechara, Mariusz Nowacki

Chromosome (SMC) proteins are a large family of ATPases that play important roles in the organization and dynamics of chromatin. They are central regulators of chromosome dynamics and the core component of condensin. DNA elimination during zygotic somatic genome development is a characteristic feature of ciliated protozoa such as *Paramecium*. This process occurs after meiosis, mitosis, karyogamy, and another mitosis, which result in the formation of a new germline and somatic nuclei. The series of nuclear divisions implies an important role of SMC proteins in *Paramecium* sexual development. The relationship between DNA elimination and SMC has not yet been described. Here, we applied RNA interference, genome sequencing, mRNA sequencing, immunofluorescence, and mass spectrometry to investigate the roles of SMC components in DNA elimination. Our results show that SMC4-2 is required for genome rearrangement, whereas SMC4-1 is not. Functional diversification of SMC4 in Paramecium led to a formation of two paralogues where SMC4-2 acquired a novel, development-specific function and differs from SMC4-1. Moreover, our study suggests a competitive relationship between these two proteins.

## Introduction

*Paramecium tetraurelia* (*P. tetrauelia*) is a widely distributed fresh water unicellular eukaryote and a member of the highly diverse ciliate phylum. Similar to other ciliates, nuclear dimorphism makes Paramecium a valuable model to explore genome dynamics and RNA-mediated epigenetic processes that accompany the process of the germline–soma genome differentiation. The germline micronucleus (MIC) serves as a storage of information that is passed over to the next sexual generation. The somatic macronucleus (MAC) is responsible for gene expression. The somatic genome develops from a copy of the MIC via a series of rounds of chromosome replication and large-scale elimination of DNA segments comprising of transposable elements and short transposon remnants called internal eliminated sequences (IES).

Paramecium sexual process begins with meiosis of the micronuclei, followed by mitosis of one of the haploid MICs and karyogamy (either self-fertilization or MIC exchange and fusion between two mating cells). Next, the zygotic nucleus divides twice mitotically which leads to a formation of four diploid nuclei. Two of them develop into new MACs, whereas the other two remain unchanged. This is followed by mitosis of the MICs and cell division, where each daughter cell receives two MICs and one MAC. At the same time, the old parental MAC disintegrates. At this stage, the sexual process is completed, and the cells enter the so-called vegetative stage, dividing by binary fissions (Sandoval et al, 2014).

DNA elimination during MAC development involves excision of transposable elements and 45,000 single-copy IESs, all flanked by TA dinucleotides (Bhullar et al, 2018). Most of the eliminated DNA segments reside within genes and require a very precise excision mechanism. Two sets of small RNAs defined as "scan" RNAs (scnRNAs) (Lepere et al, 2009) and iesRNAs (Sandoval et al, 2014) are involved in DNA excision during *P. tetrauelia* development. During meiosis, the MIC genome is bidirectionally transcribed and the resulting dsRNA is processed by a pair of Dicer-like enzymes Dcl2 and Dcl3 into 25-nt-long Ptiwi01/09-bound scnRNAs. scnRNAs are then "compared" to the somatic genome in the parental MAC. MIC genome-specific scnRNAs that did not find a match to the somatic genome are then transported to the developing MAC where they target DNA for elimination (Lepere et al, 2009). Next, the excised IESs concatenate and/or circularize to serve as a template for iesRNA precursors cleaved by Dcl5 into 21–30-nt-long duplexes bound by Ptiwi10/11. The newly generated iesRNAs assure a complete elimination of all transposons and IESs (Sandoval et al, 2014; Allen et al, 2017).

Condensin is a large protein complex that plays critical roles in chromosome structure and segregation during mitosis and meiosis (Hirano, 2016). Most eukaryotes have two different types of condensin, condensins I and II. Condensin has a conserved dimer structure of two large proteins involved in structural maintenance of chromosomes (SMC), SMC2 and SMC4, which are involved in organizing the genome by using the energy from ATP hydrolysis (Wood et al, 2010). The two ATPase domains cause SMC2 and SMC4 to self-fold into a head-to-tail shape. The hinge domain is located in the middle of this V-shape structure, and the antiparallel coiled-

Institute of Cell Biology, University of Bern, Bern, Switzerland

Correspondence: mariusz.nowacki@unibe.ch

coil domains connect the ATPase domains and the hinge domain. Dimerized by the hinge domains from SMC2 and SMC4, the core of the condensin forms (Anderson et al, 2002). In human (Ono et al, 2004) and other eukaryotes (Chan et al, 2004; Shintomi & Hirano, 2011), condensins I and II regulate chromosome assembly and segregation differently in both meiosis (Nishide & Hirano, 2014) and mitosis (Green et al, 2012). The condensin complex connects to chromatin and progressively extrudes a DNA loop by binding the DNA on one side and reeling from another side (Ganji et al, 2018). Furthermore, chromosomes arranged in nested loop arrays winding around a helical "spiral staircase" inside a cylindrical chromatid by condensins I and II reveal the conformation of mitotic chromosomes (Gibcus et al, 2018). In addition to condensing chromatin, condensin have been implicated in various other process, such as single-strand DNA (ssDNA) binding preference (Sakai et al, 2003), reannealing complementary ssDNA (Sutani & Yanagida, 1997), and removing ssDNA-binding proteins (Akai et al, 2011).

In ciliates such as *Tetrahymena* and *Paramecium*, chromatin state appears to play an important role in DNA elimination process. In *Tetrahymena* for instance, it is widely accepted that IES DNA is eliminated in a form of compact heterochromatic units (Madireddi et al, 1996). Specifically, a unique condensin form, condensin D, was reported to be required in the somatic nuclear maturation in *Tetrahymena thermophila* (Howard-Till et al, 2019). More recently, it has been suggested that, in *Paramecium*, transposons and IESs must be made accessible for excision through nucleosome depletion (Singh et al, 2022). Although local chromatin state is clearly associated with DNA elimination, there are no data available on the possible role of large-scale chromosome structure in this process. In a related ciliate species, *Tetrahymena*, a condensin component SMC4 has been reported to be required for amitotic MAC division (Cervantes et al, 2006) (polyploid MACs in ciliates divide without spindle formation, each daughter nucleus receives roughly half of the DNA); however, there are no reports so far that would implicate SMC4 in developmental genome rearrangement. Several *Paramecium* proteins that are required for DNA elimination, such as Dcl5 or E(z)l, form distinct foci in the developing MAC, indicating a possibility that DNA excision happens in specific small compartments within the nucleus. If that was indeed the case, the IES-containing DNA genome could have been organized in a specific way which would allow groups of IESs to be brought together before excision. Here, condensin could be involved in grouping IES-rich DNA into distinct foci.

This work focuses mainly on two paralogues of SMC4 in *Paramecium*, which have very different expression profiles, SMC4-1 is ubiquitously expressed in macro and micronuclei throughout the entire life cycle and its expression peaks during meiosis, whereas SMC4-2 protein is absent during vegetative divisions but is highly expressed during MAC development according to the expression pattern. Our analysis revealed that SMC4-2 protein is present only in the developing MAC and its silencing strongly inhibits DNA elimination and, as a consequence, the survival of the sexual progeny. We characterize the effects of SMC4-2 in more detail and that a proper global chromosome structure is essential for the process of DNA elimination.

The relationship between SMC function and IES excision in *P. tetraurelia* is supposed to be important because we assume that a developmental-specific condensin subunit acquired a unique function in Paramecium and the compacted DNA structure should transiently loosen when an IES inside the DNA loop is being excised. It has been stated that the steric hindrance generated by condensin could impact inhibit the transcription. As mentioned above, massive transcription is needed for the IES excision. The hypothesis that IESs cannot be eliminated if the corresponding DNA region was compacted into a dense chromosome could be reasonable. In the present research, we assessed the effects of two additional ohnologs of SMC4 (SMC4-1, SMC4-2), and SMC2 (SMC2-1, SMC2-2) in *P. tetraurelia*. Strong IES elimination failure is observed after RNA interference (RNAi) of SMC4-2, SMC2-1, and SMC2-2. It indicates a direct or indirect effect of condensin proteins on IES excision. SMC location indicates that SMC4-1 participates in the entire development of *P. tetraurelia*, whereas SMC4-2 primarily exhibits in the late stage. Interacting proteins of SMC4-1 or SMC4-2 have also been investigated to shed light on the functional complex of SMC4s in *P. tetraurelia*. Therefore, this study makes a contribution to research on SMC4s in *P. tetraurelia* by demonstrating the functional difference between them in IES excision.

## Results

### Identification of two SMC4 homologues in *P. tetrauelia*

In Fig 1A, the amino acid sequences of the two SMC4 homologues (SMC4-1 and SMC4-2) in *P. tetrauelia* differ significantly except for the conserved domain architecture (Fig 1B). This suggests that the presence of the two paralogs is not simply a result of a recent gene duplication where both genes have a redundant function. This raises a question of why *P. tetrauelia* needs two SMC4s. In Fig 1C, we show maximum likelihood phylogenetic tree of SMC4s from 22 species. SMC4-1 in *P. tetraurelia* is located in an independent branch, whereas SMC4-2 is closer to *Tetrahymena*, a close relative of *P. tetrauelia*. Also, *Paramecium* seems to be unique among ciliates in having two SMC4 paralogs. To compare expression differences between SMC4-1 and SMC4-2, an expression curve was made based on data from the paramecium gene expression database, shown in Fig 1D. SMC4-1 expressed continuously throughout the entire life cycle, including all developmental stages, although the expression is differentially regulated. SMC4-2 expression is restricted to postmeiotic developmental stages.

### SMC4-2 is required for DNA elimination

To determine the functions of SMC4-1 and SMC4-2 in DNA excision, RNAi was used to knock down (KD) the expression of each SMC4 at the mRNA level. The IES PCR results obtained from the preliminary analysis of the KD are shown in Fig 1E, where the retention of IES excision is shown as slow-migrating bands which represent non-excised IESs. SMC4-2 KD induces retention of all tested IES, although there was no evidence that the SMC4-1 KD has an influence on this process. We also tested SMC2-1 KD and SMC2-2 KD cells by IES PCR, but like SMC4-1 KD, no IES retention was observed. Although the efficiency of RNAi could be low, we repeated the knock down

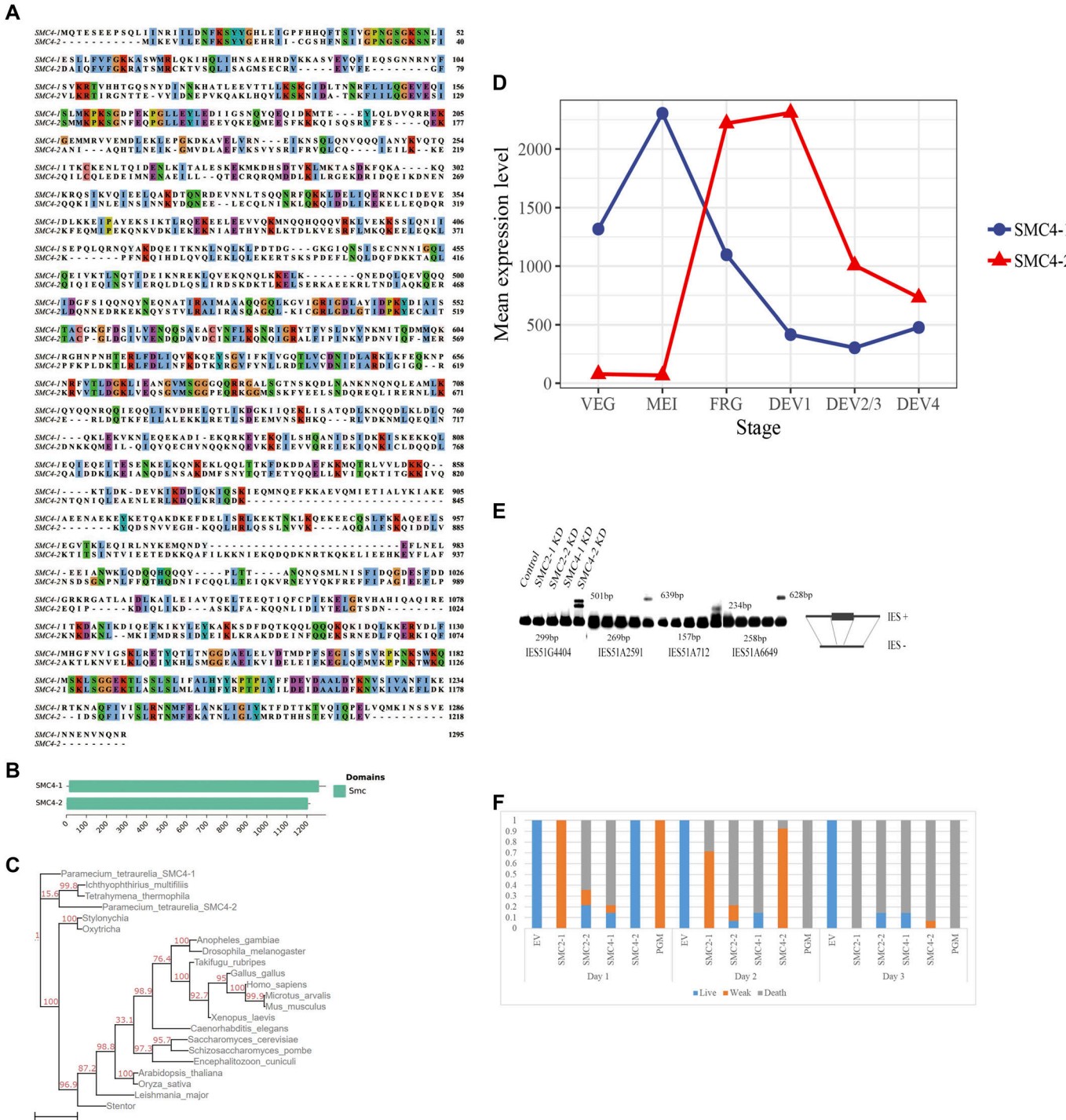

**Figure 1. Structure and function comparison between SMC4-1 and SMC4-2 in *P. tetrauelia*.**

**(A)** Protein sequence alignment between SMC4-1 and SMC4-2. The consensus amino acids marked in the same color between two sequences. **(B)** Conserved domain prediction of SMC4-1 and SMC4-2. Both of them belong to the SMC family. **(C)** Maximum Likelihood tree of SMC4s in ciliates and randomly selected organisms. **(D)** Autogamy time-course of gene expression for *P. tetraurelia* SMC4-1 and SMC4-2. The curve in blue represents the expression of SMC4-1 and the orange one indicates the expression of SMC4-2. The Y axis means the mean expression level. VEG, vegetative cells; MEI: beginning of macronucleus (MAC) fragmentation and micronucleus meiosis; FRG: population in which about 50% of cells have a fragmented old MAC; DEV1: earliest stage when a significant proportion of cells has visible MAC anlagen; DEV2/3: most of the cells with MAC anlagen; DEV4: most of the cells with MAC anlage (Arnaiz et al, 2017). **(E)** Effect of EV, SMC2-1, SMC2-2, SMC4-1, SMC4-2 knockdowns on internal eliminated sequence (IES) excision. IES retention was tested by PCR using primers flanking IES sequences. The excised form is shown as (IES−) and the unexcised form is shown as (IES+). The IES− form is always detectable because of presence of the parental MAC in the sample. The IES+ is only present in IES retention in the newly developing MAC. **(F)** Effect of EV, SMC2-1, SMC2-2, SMC4-1, SMC4-2 and PiggyMac KD on cell survival. The healthy, weak, and dead cells are shown in different colors.

experiments several times and all attempts returned the same results. A survival test is a common method for determining whether gene silencing during development had an effect on the sexual progeny's survival. As shown in Fig 1F, death rates greater than 80% were shown in all SMC KD experiments, which means that all SMCs are essential for cell survival.

For genome-wide information on how DNA elimination was affected by SMC4-2 KD, developing MACs were isolated from post-autogamous cells, whose growth was arrested by starvation to prevent the cells from dying, and high-throughput MAC sequencing analysis was performed. The distribution of IES retention scores (IRS) is shown in Fig 2A. The most striking result to emerge from this data is that 99% of IESs are affected by SMC4-2 KD, and the peak retention score is ~0.4 (i.e., on average 40% copies of each IES was retained in the new MAC), which indicates that functional SMC4-2 is required for the process of DNA elimination. This result also shows that all IES types are sensitive to SMC4-2 KD and is only PiggyMac (PGM) KD which is essential for DNA cleavage at IES ends and usually serves as a positive control in IES retention experiments shows a stronger right-skewed distribution of retained IESs (Fig 2B). The SMC4-2 KD effect is one of the most severe effects compared with other previously examined factors involved in DNA excision process. IES retention correlational analysis was performed to determine whether SMC4-2 KD's effect on DNA retention correlates with some other known proteins that are involved in IES excision (Fig 2C). Usually if two or more proteins are part of the same pathways, such as small RNA programming of excision, or chromatin factors involved in DNA elimination, their silencing tend to have a similar effect on the excision of different subsets of IESs. This leads to a high IRS correlation between these functionally related factors. In the case of SMC4-2, no such a correlation could be identified. SMC4-2 seems to not be involved in the RNA-mediated programming of DNA elimination (no correlation with Dcl2/3/5-KD, Dcl5-KD, Nowa1/2-KD, Ptiwi01/09 KD) nor in chromatin-associated DNA elimination targeting (no correlation with ISWI-KD, EZL1-KD). SMC4-2-KD does not correlate either with meiosis-specific factors that were previously shown to be essential for DNA elimination (MSH4-KD, MSH5-KD, Spo11-KD).

### Localization of two SMC4s shows diverse patterns during the cell cycle

To further investigate the developmental role of SMC4-2, we performed a GFP-fusion localization of SMC4 proteins. SMC4-2-GFP vector containing 352-bp upstream and 267-bp downstream regions, in addition to N-terminal fusion of GFP and SMC4-2 was microinjected it into the vegetative MACs. The localization of SMC4-2-GFP in Fig 3A shows that the fusion protein is present exclusively in newly developing MACs and it disappears before the first post-development cell division. The place and time of the protein expression and localization coincided with DNA elimination process.

This experiments further supports the hypothesis that SMC4-2 is not involved in any sort of cell division (meiosis or mitosis). SMC4-1-GFP, however, appears in the vegetative MAC, also during the amitotic MAC divisions, and in the micronuclei (Fig 3B). During development, SMC4-1-GFP remains present in the meiotic and postmeiotic micronuclei and it appears in the developing MACs, but at the same time, the signal in the parental MAC fades away. Both SMC4-1-GFP and SMC4-2-GFP appear granular in the developing MACs. This suggests that the proteins are not homogeneously distributed in the nuclei. A similar granular pattern can be seen for SMC4-1-GFP in the vegetative MAC.

### Knockdown of SMC4-2 affects gene expression levels

One of the roles of condensin is establishing and maintaining a 3D chromosome structure within the nucleus (Hirano, 2012). It is widely accepted that the exact position of DNA has an influence on gene expression levels which can be easily tested experimentally by modifying the chromosome 3D structure within the nucleus (Gibcus & Dekker, 2013). However, a more recent study on *Drosophila* suggests that 3D chromosome structure is not associated with gene expression levels (Ghavi-Helm et al, 2019). To test whether the effect of SMC4-2 KD on DNA elimination could be because of the abnormal expression of genes that may play a role in genome rearrangement, we performed mRNA sequencing of samples from late time of EV controls and SMC4-2 KD. Fig 4A shows that SMC4-2 KD affects the expression level of a subset of genes. In *Paramecium*, most of the genes required during sexual development are expressed from the parental MAC. Some of them, however, are known to be expressed from the developing MAC. For example, PTIWI10 and PTIWI11, which bind iesRNAs, are transcribed in the new MAC after IESs located in their promoters are excised. In our SMC4-2 KD experiment, both genes were down-regulated (Fig 4B). This effect, however, may be indirect because of the retention of IESs that are located within PTIWI10/11 transcription promoters. Several other genes affected in our experiment were up-regulated (Fig 4B), which would support the idea that gene expression may be directly linked to chromosome 3D structure.

### SMC4-2 knockdown disrupts the production of iesRNAs

Because SMC4-2 KD affects developmental DNA excision and also the expression of the iesRNA pathway genes PTIWI10 and PTIWI11, we wished to take a closer look at development-specific small RNA populations in the SMC4-2–silenced cells. Normally, iesRNAs help to complete DNA elimination and are not essential for the survival of sexual progeny (Sandoval et al, 2014). In the absence of iesRNAs, most of the IESs are excised in the new MAC because the bulk of excision happens in an RNA-independent manner, and the RNA-dependent IES excision requires primarily scnRNAs. In the case of

---

Blue: percentage of healthy cells (cell growing at a normal rate); orange: percentage of sick cells (altered number of divisions or behavior); grey: dead cells. PiggyMac KD was positive control. Empty L4440 vector was negative control. Day 1, Day 2, Day 3: days post autogamy. Single post-autogamous cell was picked into new individual culture and observed the survival in consecutive 3 d. For each bar n = 14 cells.

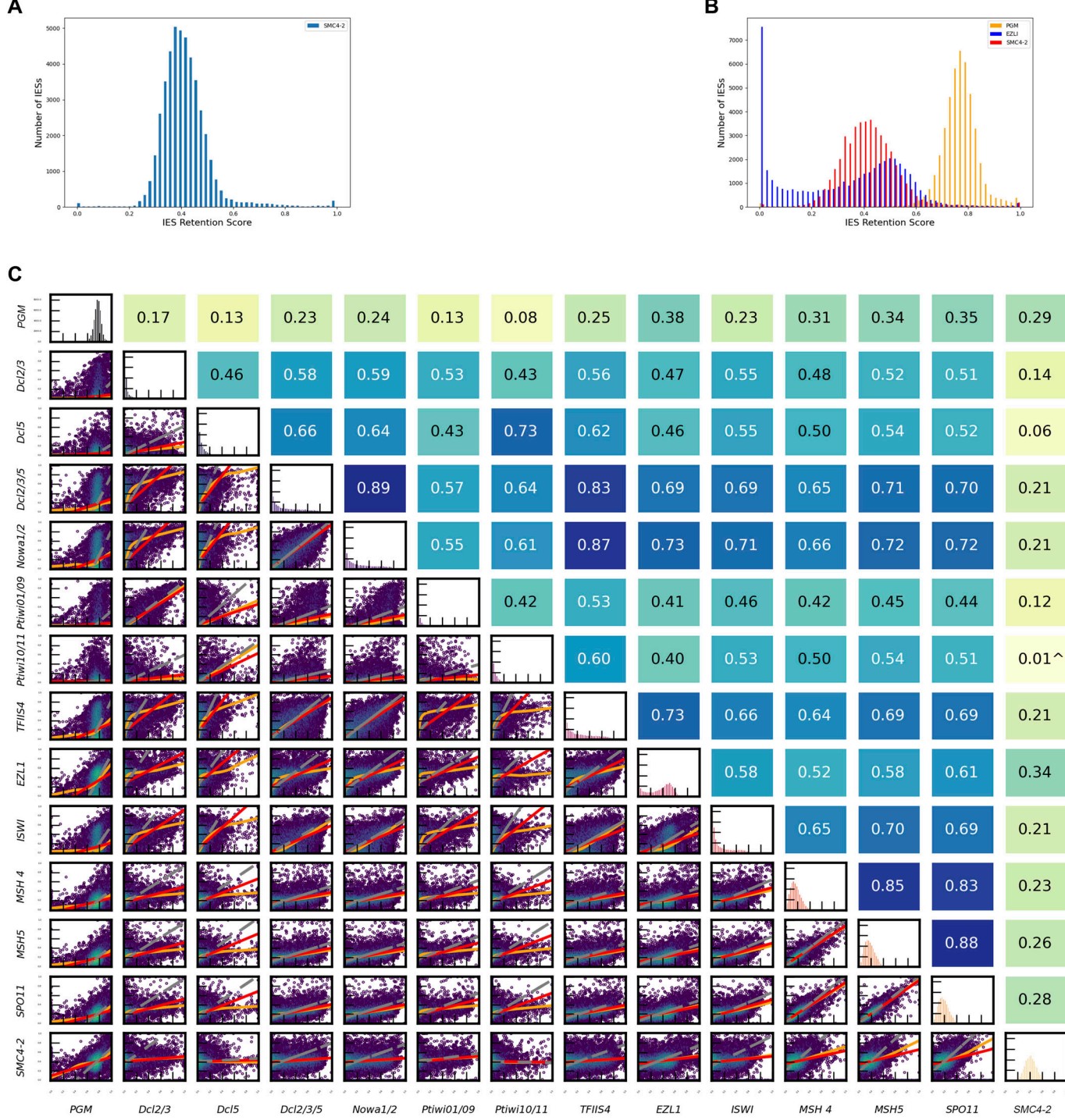

**Figure 2. Internal eliminated sequences (IES) retention pattern analysis.**
**(A)** Retention score distribution determined by sequencing DNA extracted from a cell fraction enriched in a new macronucleus after SMC4-2 KD. Y axis represents the number of IES. X axis means the percentage of single unexcised IES among its copies. **(B)** Integrated IES retention distribution of PiggyMac (in orange), EZL1 (in blue), and SMC4-2 (in red) KDs. **(C)** Relationships in IES retention. The retention score of PiggyMac, DCL2/3, DCL5, DCL2/3/5, NOWA1/2a, Ptiwi01/09, Ptiwi10/11, EZL1, TFIIS4, ISWI are from our laboratory, SMC4-2 retention score is from this study. Correlation coefficients of Pearson are given at the right part of the graph correspondingly. Red lines represent ordinary least-squares regression, orange lines are for LOWESS, and gray lines indicate orthogonal distance regression.

SMC4-2 KD, the effect on DNA elimination is much more profound and it is highly unlikely that it could be only because of the disruption of the iesRNA pathway. SMC4-2 KD affects the excision of both RNA-dependent and RNA-independent IESs. To determine the effect of SMC4-2 KD on small RNAs we performed a high-throughput sequencing of short RNAs purified during development. The result

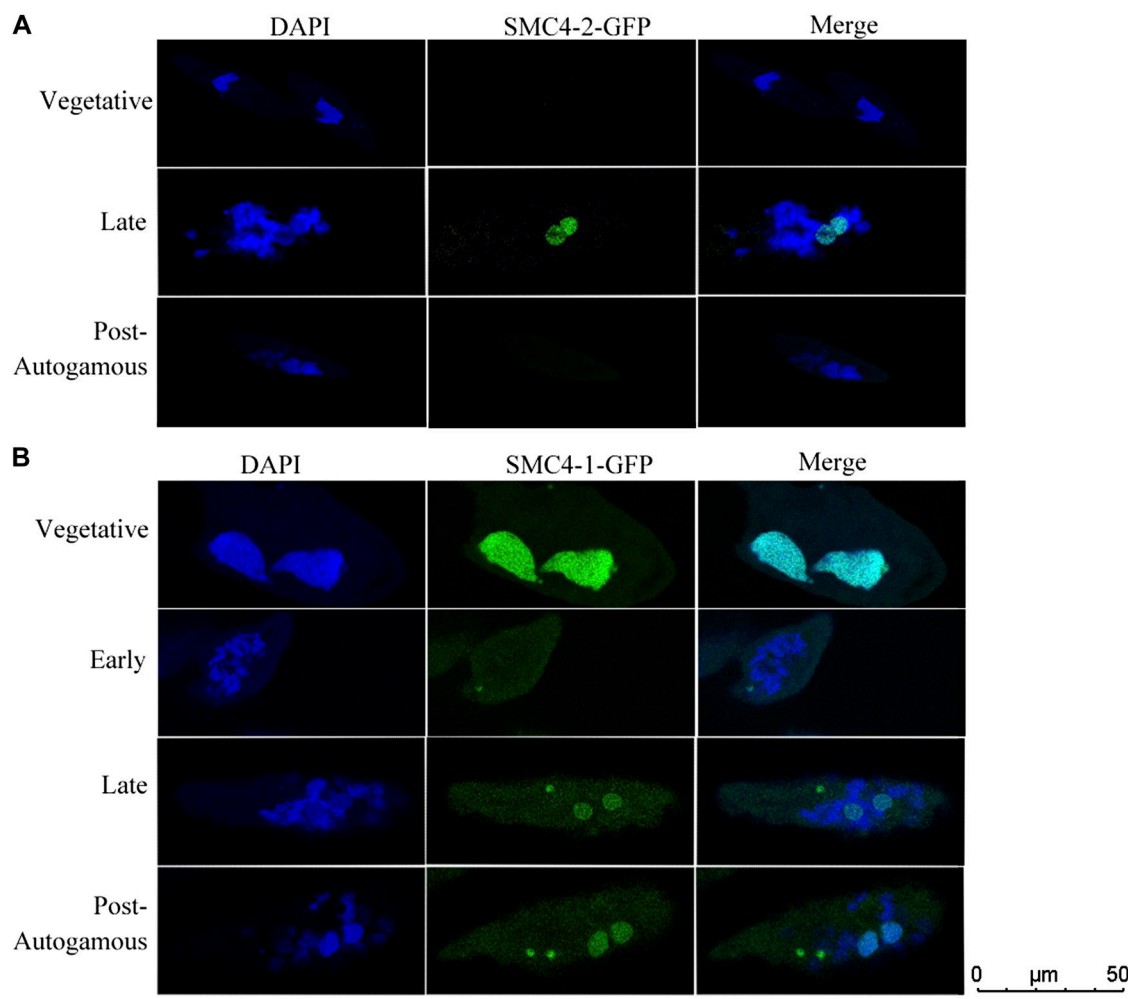

**Figure 3. Localization of SMC4-1 and SMC4-2 tagged with GFP.**
**(A)** SMC4-2 tagged with GFP was exclusively localized in the macronuclear. **(B)** SMC4-1 tagged with GFP has a wide distribution including vegetative macronuclear and micronuclear, new developing macronuclear and micronuclear, not in macronuclear which is going to be fragmented. DAPI staining in blue represent DNA. Vegetative: vegetative cells; Early: 20% cell shows fragmented old macronucleus (MAC); Late: 2–4 h post 100% cell exhibits fragmented old MAC, new developing MAC starts to be observed. Post-autogamous: autogamy finishes, cell division, ready for the next vegetative growth.

shows no effect of SMC4-2 KD on scnRNAs (25 nt long); however, iesRNAs were completely gone (see the absence of red bars in 26–28 nt RNAs in Fig 5B) compared with the negative control (Fig 5A). Because iesRNAs are produced from the excised portion of the genome and SMC4-2 KD leads to the retention of nearly every IESs, but at the average level of 40% (40% of the 800 copies of each IES), the remaining 60% of IESs that are excised would normally still produce iesRNAs. In SMC4-2 KD, however, almost no iesRNAs are present, which suggests that the reason for this absence is most likely the low level of PTIWI10/11 transcription. This result also tells us that the effect of SMC4-2 KD on IES excision is not (even partially) because of an absence of scnRNAs.

### Co-silencing of SMC4-2 with other subunits reverts IES retention phenotype

The above-mentioned experiments show that SMC4-2 silencing affects *Paramecium* genome rearrangement but it is not entirely

clear whether this phenotype is because of the absence of SMC4-2-containing condensin complex or whether SMC4-2 functions in a condensin-independent manner. Although this is a very unlikely possibility, we decided to test it. Double silencing of interacting proteins would, in theory, return enhanced or at least similar retention phenotype of each single silencing. We performed double silencing of SMC4-2 with SMC2-1, SMC2-2 or with SMC4-1. Surprisingly, the IES retention phenotype disappeared in all these double-silenced cells (Fig 6A). The cell survival test in Fig 6B shows all the double knockdowns had a lethal effect on sexual progenies. This effect was confirmed by using three separate silencing constructs, each one containing two target genes. This way, each silencing construct produces equal amounts of dsRNA for each pair of targets. According to previous single-silencing results and as shown in Fig S1, introducing dsRNA of SMC2-1, SMC2-2 or SMC4-1 induced morphological changes in the MAC, and the cell number before autogamy dropped dramatically. We therefore induced autogamy by transferring the cells to a bacteria-free medium before their

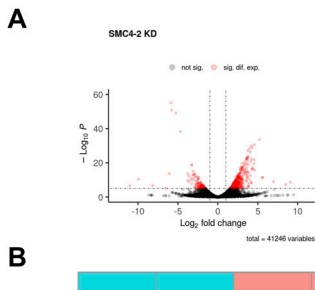

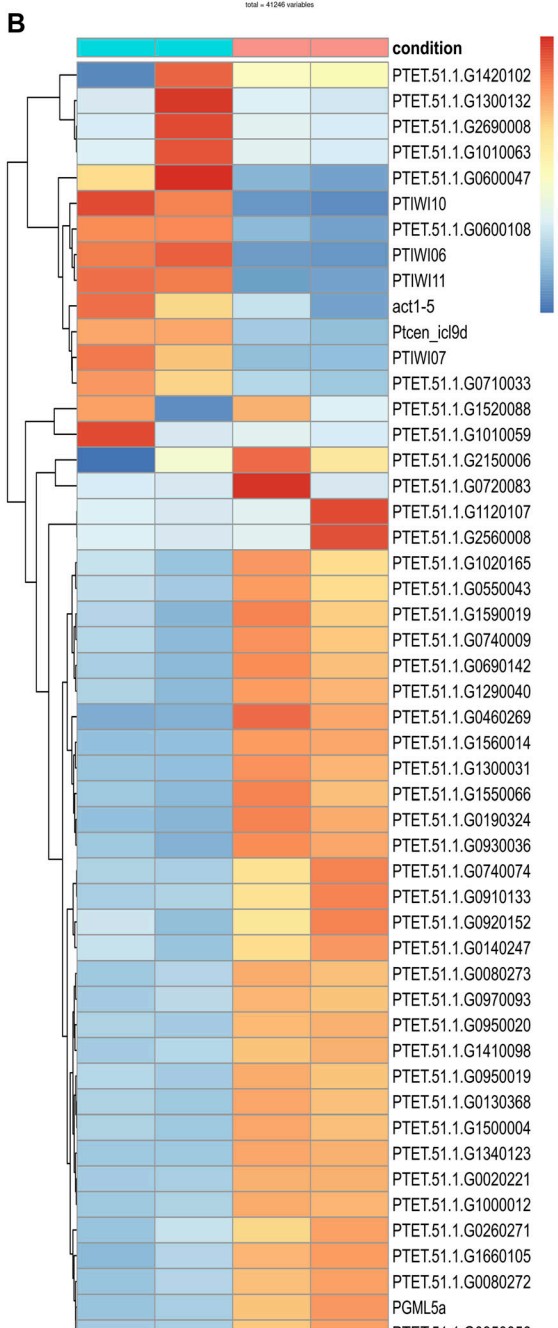

**Figure 4. Differential expression in a SMC4-2 KD.**
**(A)** Volcano plots showing the differentially expressed genes detected in SMC4-2 KD compared with control. The most statistically significant genes are shown toward the top, with up-regulated genes on the right and down-regulated proteins on the left. Gray dots mean no significant difference, green dots represent

number decreased. Also in this case, the silencing was lethal for the sexual progeny, with no obvious IES retention phenotype. But the co-silencing phenotype and the survival test could also because of the reduced silencing efficiency when knocking down SMC4-2 with SMC2-1 or SMC2-2 or SMC4-1. Fig 6C shows results from the analysis of single- and double-silencing experiments. Taken together, these results support the possibility that there is an unusual relationship between SMC4-2 and SMC2-1/SMC2-2/SMC4-1.

### Co-immunoprecipitation (Co-IP) of interacting partners of SMC4-1 and SMC4-2

To identify proteins interacting with SMC4-1 and SMC4-2, we performed Co-IP followed by mass spectrometry. We expressed Flag-HA–tagged SMC4-1 and SMC4-2 (Fig 7A), and both were detected by an anti-HA primary antibody in the cell lysate and in the column-bound fraction as arrow heads point. Analysis by shotgun liquid chromatography tandem mass spectrometry (LC-MS/MS) of non-labelled samples (Fig 7B) revealed differences in proteins interacting with SMC4-1 and SMC4-2. There was no evidence that PGM or other proteins known to be involved in genome rearrangements interact directly with. Possible reasons include transient interacting with SMC4s, low efficiency under non-crosslinking, or a novel mechanism of IES excision which do not need those proteins participation. A more surprising observation is the enrichment difference of SMC2 comparison with SMC4-1 and SMC4-2. Venn diagrams (Fig 7B and C) illustrate the relationships of differential enrichment and unique proteins among co-purified with SMCs. In Fig 7B, the grey arrow points to SMC2-1, indicating that it is highly enriched in both SMC4-1 and SMC4-2 Co-IP experiments compared with control, but there was no difference in enrichment between SMC4-1 and SMC4-2. This indicates that SMC4-1 and SMC4-2 may bind the same amount of SMC2-1 at this time point. The red arrow points to SMC4-2 and SMC2-2 indicate these two proteins were not only enriched in SMC4-1 and SMC4-2 compared with control but were also enriched in SMC4-2 compared with SMC4-1. The blue arrow points to SMC4-1 in Fig 7C, indicating that SMC4-1 is a unique protein that can only be seen in the SMC4-1 interacting dataset. Together, these results provide completely new insights into the diverse elements of SMC4-1 and SMC4-2 interactions in the development of *P. tetrauelia*.

## Discussion

Condensin is evolutionarily conserved protein complex responsible for chromosome maintenance in nearly all organisms (Hirano, 2016; Uhlmann, 2016). It plays a central role in DNA

significant only in fold change. The X-axis represents $\log_2$ (fold change) values and Y-axis represents $-\log_{10}$ (P-val) values. **(B)** The top 50 most differentially expressed genes in SMC4-2 KD in comparison with the EV. Light blue on the top represents two replicates of EV, the orange means two replicates of SMC4-2 KD. The red and blue colors and intensity of the boxes represent changes of gene expression based on z-transformed normalized read counts generated by DESeq2.

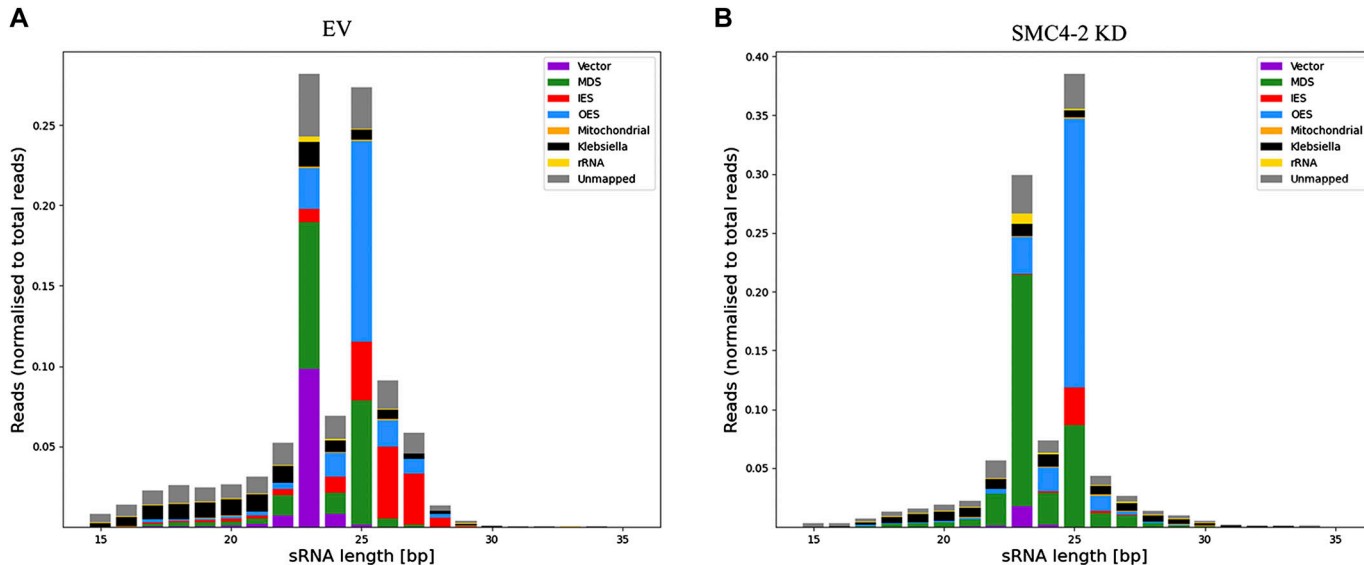

**Figure 5. Small RNA sequencing in EV and SMC4-2 KD.**
**(A, B)** Histograms of small RNAs classified by length. Both panels show the distribution at late time point (specifically 4 h after 100% fragments). The reads map to the macronucleus genome is shown in green or internal eliminated sequences in red. A significant decrease of 26–30 bp, in which the iesRNAs are, can be found in SMC4-2 KD.

replication (Terakawa et al, 2017), chromosome condensation and segregation (Hirano et al, 1997), gene expression (Wood et al, 2010), and DNA damage repair (De Piccoli et al, 2009). In Paramecium, the germline micronuclei generally resemble classical eukaryotic nuclei that divide mitotically and meiotically. Development of a new MAC is a process where the germline nucleus and germline genome undergoes a multistep transition which includes a massive DNA endoreplication and DNA elimination. In this work, we show that the development of a new MAC requires a uniquely dedicated condensin subunit—SMC4-2. This protein is not only unique in its localization and expression pattern (specific for new MAC development), but also in its function, as it is required for DNA elimination. It has been reported previously that another ciliate, Tetrahymena, uses a development-specific condensin which is required for DNA endoreplication and IES excision (Howard-Till et al, 2019).

If the primary function of Paramecium SMC4-2 is chromosome condensation, our results may suggest that DNA elimination process is dependent on a specific DNA condensation state. Our previous data show that DNA elimination is dependent on local chromatin state changes but this only affects a relatively small subset of IESs and transposons—specifically those located in heterochromatic regions (Singh et al, 2022). SMC4-2 knockdown, however, seems to be affecting DNA elimination globally. Virtually all of the IES excision is affected by the absence of SMC4-2. This suggests that the mechanism of SMC4-2–dependent DNA elimination is different from the previously reported ones.

Because other members of the Paramecium SMC family, including SMC2s, SMC4-1, SMC1, and SMC3, do not show a similar function, we cannot exclude a possibility that SMC4-2 has a unique function in addition to serving as the expected condensin component. The mechanism by which SMC4-2 regulates DNA elimination is still unknown, and from our structural prediction, no significant

characteristics can be presumed to be the reason for this regulation. Therefore, indirect regulation may be considered.

Another interesting observation is that SMC4-2 is localized specifically in newly developing MACs, whereas SMC4-1 localizes to all the nuclei (MIC, MAC, and developing MAC). If SMC4-1 plays a classical role as a condensin subunit and is involved in chromosome maintenance and nuclear division, SMC4-2 may play an "unorthodox" role that is limited to macronuclear development. Indeed, SMC4-1-GFP expression shows that SMC4-1 may be the canonical condensin subunit as the GFP signal was present throughout the entire life cycle in both types of nuclei. This result is consistent with data obtained for Tetrahymena, which suggests that Smc4p may be involved in both amitotic and mitotic nuclear divisions (Cervantes et al, 2006). One unanticipated finding was that the SMC4-1-GFP signal was not present in old MAC fragments, which may suggest a dynamic dissociation of SMC4-1 from the genome as cells undergo autogamy. This may be related to the "genome scanning" process, when MIC-produced small RNAs interact with long noncoding transcripts produced from the entire "old" MAC genomes (Michelini et al, 2018). The massive transcription events correlate with the brief absence of SMC4-1. Otherwise, our colocalization experiment confirms the temporal and spatial association between SMC4-1 and SMC4-2 at subsequent developmental stages.

A possible explanation for the effect of SMC4-2 knockdown on DNA elimination is that it affects the expression of genes involved in this process. Because SMC4-2 is present only in the developing MAC, it would be reasonable to assume that those affected genes are normally expressed from the developing MAC. However, most of the factors shown previously to be involved in developmental genome rearrangement are expressed from the old MAC. One exception is Ptiwi10/11, a pair of paralogs that is transcribed in the developing MAC, right after an IES located within their promoters is excised (Furrer et al, 2017). Because the global IES retention post SMC4-2

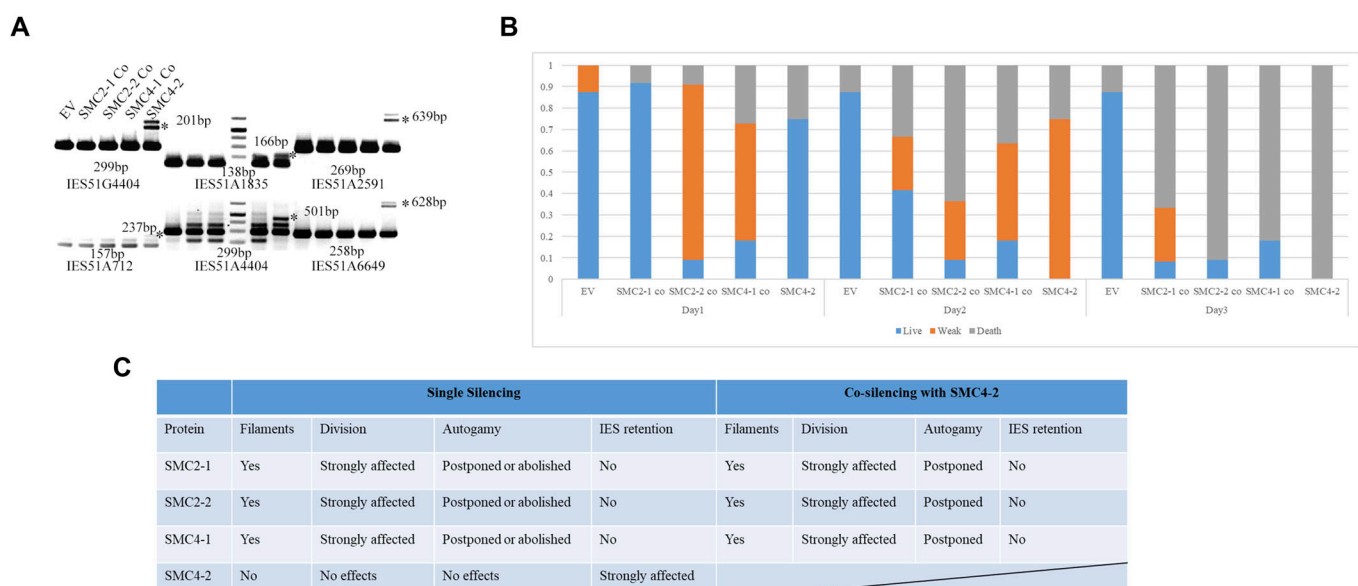

**Figure 6. Co-silencing of SMC4-2 with SMC2-1, SMC2-2 or SMC4-1.**
**(A)** Internal eliminated sequences (IES) PCR of co-silencing. IES retention was tested by PCR using primers flanking IES sequences. The asterisk indicates the retained IESs. **(B)** Survival test of co-silencing. **(C)** Summary of single and co-silencing effect on nuclear morphology, autogamy process, and IES retention. Filaments represent the filaments between two macronuclear in dividing cells (as shown in Fig S1). Division affected was drawn from the unsuccessful dividing cell. Autogamy postponed was indicated because the delayed start of autogamy after RNAi compared with control.

silencing is entirely different from the effect of Ptiwi10/11 knock-down, we do not believe that the effect is because of the absence of Ptiwi10/11 mRNAs. Our mRNA sequencing after SMC4-2 knock-down shows that Ptiwi10/11 transcripts were indeed strongly down-regulated (Fig 4B), which is not surprising as Ptiwi10/11 expression requires prior IES excision. Therefore, we conclude that this effect on transcription is an indirect consequence of DNA elimination impairment by SMC4-2 knockdown. Similarly, the mRNA sequencing showed several other transcripts being affected (Fig 4B), and just like Ptiwi10/11, these genes may need to be properly rearranged before they can be correctly expressed. Another interesting effect of SMC4-2 KD is the up-regulation of a number of genes. It is possible that in some cases, non excised IESs might act as transcription promoter, such as mtA gene (Singh et al, 2014).

The abovementioned effect on Ptiwi10/11 expression should, as a consequence, affect the production of iesRNAs, a class of small RNAs that is produced from excised DNA and acts as a positive feedback to ensure complete DNA elimination (Sandoval et al, 2014; Allen et al, 2017). Because Ptiwi10/11 is involved in the iesRNA pathway through binding iesRNAs, one should expect that the down-regulation of Ptiwi10/11 expression would reduce the amount of iesRNA. In addition, because iesRNAs are produced from the transcripts of excised IESs, even in the presence of Ptiwi10/11, iesRNA level is expected to be low. Indeed, our small RNA sequencing indeed shows a complete absence of iesRNAs (Fig 5B, absence of the red bars in 26–28 nt sRNAs).

Our double-silencing results presented here (Fig 6) lead to an unexpected outcome of restoring DNA elimination which could because of the low knock-down efficiency. In another aspect, if SMC4-2 is a part of condensin complex together with SMC2s, a double silencing (SMC2/4-KD) should, in theory, either enhance the

IES retention phenotype or at least keep it at the same level. If SMC4-2 does not interact with SMC4-1 and SMC2s, then, in theory, the double silencing experiments should exhibit combined phenotypes of the single silencing (IES retention and morphological changes). In our experiment, only the latter phenotype was present in the double silencing. IES elimination remained unaffected. It is difficult to explain this result.

The LC-MS/MS results provide detailed information on SMC4-1 and SMC4-2 interactions with other proteins. We found that no proteins known to be involved in DNA elimination are present in the SMC4-2 interaction dataset. Similarly, no such proteins were found interacting with SMC4-1. In our Co-IP experiment with SMC4-1-Flag-HA as bait, all SMC4s and SMC2s were detected as interacting partners, but only SMC2s could be identified when using SMC4-2-Flag-HA as bait (no SMC4-1 present). This led us to reconsider the relationship between SMC4-1 and SMC4-2. If they interact with each other, then the absence of SMC4-1 in the SMC4-2 pull-down experiment does not make sense, because SMC4-2 was found interacting with SMC4-1. In these two experiments, the only thing that has been strongly affected was the gene copy number. Microinjection of a transgene introduces a large excess of the gene copy number compared with the endogenous one. Under these conditions, the quantitative ration between SMC4-1 and SMC4-2 proteins was likely either reduced or enhanced. This could explain why SMC4-1/SMC4-2 interaction has been lost. It is possible that SMC4-2 could replace SMC4-1 in the condensin complex by a competitive binding. This assumption is based on the fact that SMC4-1 and SMC4-2 share conserved domains and the potential to form a condensin complex with SMCs. A large copy number of SMC4-2 could result in a competition with endogenous SMC4-1 for binding to SMC2 proteins, and SMC4-1 might get dissociated from chromosomes but be not degraded, according to

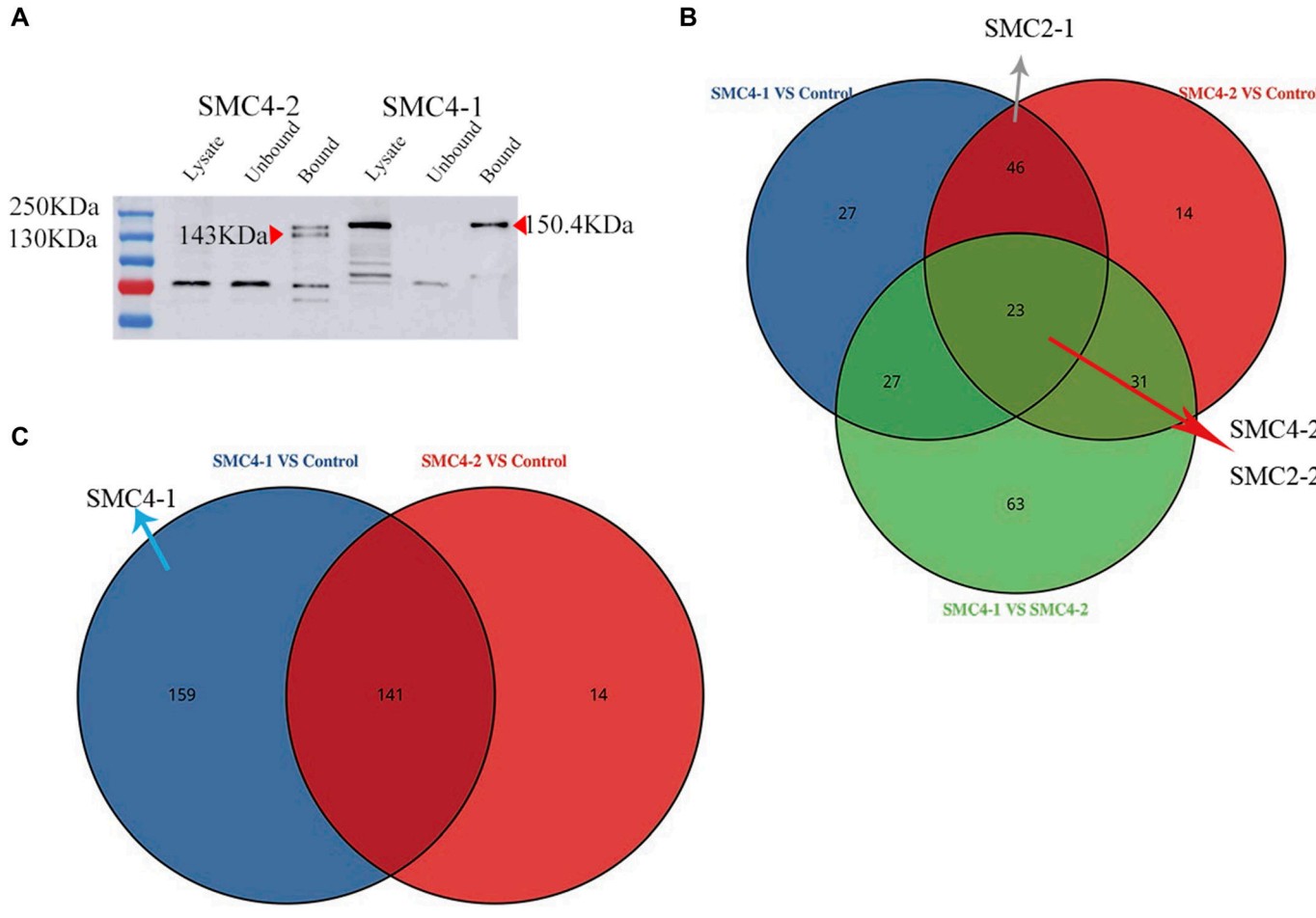

**Figure 7. Co-immunoprecipitation of Flag-HA tagged SMC4-1 and SMC4-2 at late time point.**
**(A)** Western blot of Flag-HA tagged SMC4-2 and SMC4-1 pulled down with anti-HA antibodies. Lysate: cell lysate; Unbound: supernatant by spinning down anti-HA beads-lysate mixture after overnight incubation; Bound: enriched target proteins on beads after five times washing. Molecular weight of SMC4-1 ~150KD; molecular weight of SMC4-2 ~143KD. **(B, C)** Venn diagram of differential enrichment and unique proteins versus control. **(B)** Venn diagrams showing the overlap in differential enrichment of SMC4-1 versus control, SMC4-2 versus control, and SMC4-1 versus SMC4-2. The gray arrow represents nonregulated in SMC4-1 versus SMC4-2. The red arrow means up-regulated in SMC4-2 compared with SMC4-1. **(C)** Venn diagrams depicting the overlap in unique proteins of SMC4-1 versus control and SMC4-2 versus control. The blue arrow indicates unique protein only can be seen in SMC4-1 enrichment.

the continuous GFP fluorescence. The "SMC4-2/SMC2" condensin complex could be the developing MAC-specific condensin form that is essential for DNA elimination. After injecting the SMC4-1 transgene into the parental MAC, the copy number ratio between SMC4-1 and SMC4-2 could likely be reversed, and SMC4-2 could still be able to occupy SMC4-1's position, whereas a high amount of SMC4-1 could inhibit this process.

In summary, the results presented in this study point towards a "competition" mechanism that could explain the function of SMC4-2 in Paramecium. Our data suggest that SMC4-2 might have an unorthodox role in developmental genome rearrangement, which goes beyond the classical function of SMC4 in chromosome maintenance (Hirano, 2006; Burmann et al, 2013). Despite the data implicating SMC4-2 more directly in the DNA elimination process, the exact function of this protein remains enigmatic. Also, the fact that only certain Paramecium species have two SMC4s may suggest that the possible involvement of specialized SMC4 in genome rearrangement may have appeared exclusively in this lineage.

Despite its limitations, our study suggests an interesting possibility that a diversification of evolutionarily conserved proteins may enable them to acquire more unique and specialized functions. Paramecium is an excellent model to study functional diversification of conserved molecular mechanisms because of its evolutionary history that involves several whole-genome duplication events (Aury et al, 2006).

## Materials and Methods

### Paramecium cultivation

All experiments were performed with mating type 7 of strain 51 of *P. tetraurelia*. *Klebsiella pneumoniae*-infected wheat grass powder medium (WGP; Pines International) with 0.8 mg/liter of β-sitosterol was treated to cultivate the cells (Merck). Cultivation took place at a temperature of 27°C (Beisson et al, 2010b, 2010c).

## Sequence alignment, domain prediction, and phylogenetic tree generation

The *Paramecium* SMC4-1 (PTET.51.1.P0410063) and SMC4-2 (PTET.51.1.P0590135) sequences extracted from ParameciumDB (https://paramecium.i2bc.paris-saclay.fr/). Other SMC4 protein sequences were collected from previous reports (Cobbe & Heck, 2004; Cervantes et al, 2006). Multiple-sequence alignment was carried out in MAFFT (Katoh et al, 2019). Conserved domain prediction of SMC4-1 and SMC4-2 were performed with the Conserved Domain Database in NCBI (Marchler-Bauer et al, 2011). Expression pattern build according to a previous report (Arnaiz et al, 2017). Phylogenetic tree was generated by using the W-IQ-TREE with default settings (Trifinopoulos et al, 2016).

## Single- and double-gene silencing of SMC2s and SMC4s

Knockdowns (KD) of SMC2s and SMC4s were done using RNAi by feeding with *E.coli* expressing double-strand RNAs as described before (Beisson et al, 2010d). The coding regions of the target genes were PCR amplified from genomic DNA using the primers listed in Table S1. Next, one or two target sequences were inserted between two inverted T7 promoters of the L4440 RNAi plasmid (Fire et al, 1998). The plasmids were introduced into HT115 (DE3) *Escherichia coli* feeding strain. An "empty" L4440 vector (EV) without insertion was used as a negative control. In addition, a PGM RNAi plasmid was chosen as a positive control (Baudry et al, 2009). Cross-silencing of other Paramecium genes, according to RNAi off-target tests performed using the ParameciumDB (Arnaiz & Sperling, 2011) tool, is unlikely. 200 cells/ml of *Paramecium* culture were transferred into the silencing medium. Double-silencing experiments were performed in two alternative ways: one was by seeding cells directly into a mix of two silencing media or 100 cells/ml cultures were seeded into 100 ml of double-silencing medium with *E.coli* containing a single plasmid with two silencing targets. After 24 h, cell morphology was observed under the microscope to check the silencing phenotype in vegetative cells. Then the cells were harvested and transferred into fresh silencing medium at a cell concentration that will allow shortening the time needed for the cells to starve and begin the sexual process (autogamy). 14 single cells that had successfully completed sexual reproduction in the silencing media were then isolated and transferred to a medium bacterized with klebsiella (regular food source) to assess the survival of the progeny after autogamy.

## DNA extraction, IES retention PCR, and Illumina sequencing

Total DNA from 100 ml of each postautogamous cultural was extracted with the GenElute Mammalian Genomic DNA Miniprep Kit (G1N70-1KT; Sigma-Aldrich). IES retention PCRs were analyzed using genomic DNA and certain primers as previously described (Sandoval et al, 2014). For deep sequencing, DNA of developmental MAC from 400 ml cells was extracted as previously described (Arnaiz et al, 2012). According to established Illumina techniques, a 150-cycle paired-end Illumina TruSeq DNA library was created and sequenced at the University of Bern's NGS platform.

## Calculation of IRSs in genome-wide and correlation matrix

The IRSs were estimated using ParTIES (Denby Wilkes et al, 2016). The number of reads corresponding to the excised IESs with just the MAC IES junction is denoted as IES, whereas the number of reads that accommodates the IES sequence is denoted as IES+. Only read pairs that were clearly mapped were counted. Each read was tallied just once to prevent excessive counting brought on by paralogous matches. To prevent length biases brought on by IES length variance, reads were exclusively counted at IES ends. Then, an IRS is determined as follows: IRS = $IES^+/(IES^+ + IES^-)$. Correlations were estimated with the Pearson method (Bechara et al, 2022).

## Total RNA extraction, mRNA sequencing, and small RNA (sRNA) analysis

Total RNA was obtained from 200 ml of *P. tetraurelia* at 4 h after 100% fragmentation. This timing is determined according to the strong GFP signal of SMC4-2 on Western blot. TRI reagent (Sigma-Aldrich) extraction was done following the suggested protocol. A stranded mRNA library was produced basing on standard Illumina protocols and sequenced with 100 cycles single-end at the NGS platform at the University of Bern. For sRNA sequencing, total RNA was sequenced by Fasteris SA. An Illumina sRNA-seq library was constructed according to standard Illumina protocols and sequenced with 50 cycles single-end. The sRNAs were categorized into several size sets (15–35 nts), then aligned with HiSat2 (version 2.1.0) using default parameters (Bechara et al, 2022). Those reads mapped were sorted to OES, IES, and MAC sequences, the mitochondrial, DNA from *Klebsiella pneumoniae*, and vector backbone.

## The GFP and mCherry fusion constructs, microinjection, and localization

he SMC4-1 or SMC4-2-GFP fusion construct under the endogenous regulatory sequences, respectively, contained MAC sequences upstream of the ATG and downstream of the TGA. The optimized GFP coding sequence (Nowacki et al, 2005) was inserted ahead the stop codon. Before performing the microinjection, all plasmids containing the fusion transgene were digested with the AhdI (R0584; New England BioLabs) or SapI (R0569L; New England BioLabs) to linearize them. The products were filtered through 0.22 $\mu$m Ultrafree MC GV filter (UFC30GV0S; Millipore), and precipitated with pure ethanol. Next, DNA was dissolved in DNase-free ddH$_2$O to a final concentration ~5.5 $\mu$g/$\mu$l. Finally, linearized DNA was microinjected into vegetative cell MACs (Beisson et al, 2010a). SMC4-1-mCherry constructs are as described above. Positive injections were picked up by checking green and red signals under a microscope. Positive single clone was expanded to high density (3,000 cells/ml). At various life cycle phases, small samples were taken and counterstained with DAPI. Fluorescence microscopy was used to detect GFP localization (Leica AF6000 system).

### Flag-HA fusion construct, microinjection, immunoprecipitation, mass spectrometry

Fusion construct and microinjection performed as described above. Positive injection was confirmed by Dot Blot (Furrer et al, 2017; Rzeszutek et al, 2022). Immunoprecipitation was performed as described before (Reuter et al, 2009; Hoehener et al, 2018). Non-crosslinking was performed because the IP of SMC4-2 under crosslinking did not work at pH below 10.4. In detail, 400-ml cells were harvest at 4 h after 100% fragmentation, pellets were resuspended in 2 ml fresh lysis buffer (50 mM Tris pH 8.8, 150 mM NaCl, 5 mM $MgCl_2$, 1 mM DTT, 1% Triton X-100, 1× protease inhibitor complete tablet [Roche], and 10% glycerol) and sonicated until complete lysis. The cell lysates were spin down at 13,000$g$, 4°C for 30 min 1 ml of the supernatant was incubated with 50 $\mu$l of Anti-HA affinity resin (Roche) overnight at 4°C while rotating. Another 1 ml supernatant was frozen in liquid nitrogen and store at –80°C for later using. Beads were washed with 1 ml IP buffer (10 mM Tris pH 8.8, 150 mM NaCl, 0.01% NP-40, 1 mM $MgCl_2$, 1× protease inhibitor, and 5% glycerol) for three times before incubation. After overnight incubation, beads were washed with 1 ml IP buffer for five times. Washed beads were resuspended in 50 $\mu$l IP buffer, boiled with 25 $\mu$l 5× SDS loading buffer at 95°C, after cooling down on ice, immediately used for Western blot and mass spectrometry analysis in the University of Bern.

## Data Availability

Nucleic acid sequences generated in this study were deposited in the sequence read archive (SRA) with accession number SRS17423120 (sRNA-seq), SRS17423122 (Macronuclear DNA-seq) and SRS17423121 (mRNA-seq). The mass spectrometry proteomics data have been deposited to the ProteomeXchange Consortium via the PRIDE (Perez-Riverol et al., 2022) partner repository with the dataset identifier PXD041826.

## Supplementary Information

## Acknowledgements

We thank Dr. Nassikhat Stahlberger for technical support and all members of the Nowacki Lab for input and discussions. We also thank the Proteomics and Mass Spectrometry Core Facility (PMSCF) at the Department for Biomedical Research (DBMR), University of Bern, Switzerland, for advice and mass spectrometry analyses. This work was supported by European Research Council grants (ERC) 260358 "EPIGENOME" and 681178 "G-EDIT;" Swiss National Science Foundation grants 31003A_146257, 31003A_166407, and 310030_184680; and grants from the National Center of Competence in Research (NCCR) RNA and Disease to M Nowacki, and by the China Scholarship Council (CSC) Award No. 201808620203 to F Zhang.

## Author Contributions

F Zhang: conceptualization, data curation, formal analysis, validation, investigation, visualization, methodology, and writing—original draft, review, and editing.
S Bechara: data curation, formal analysis, and methodology.
M Nowacki: conceptualization, resources, formal analysis, supervision, funding acquisition, project administration, and writing—review and editing.

## Conflict of Interest Statement

The authors declare that they have no conflict of interest.

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
