## [Reviewer comments · Life Science Alliance]

Life Science Alliance

Structural maintenance of chromosomes (SMC) proteins are required for DNA elimination in Paramecium

Fukai Zhang, Sebastian Bechara, and Mariusz Nowacki

DOI: <https://doi.org/10.26508/lsa.202302281>

Corresponding author(s): *Mariusz Nowacki, University of Bern*

Review Timeline:

Submission Date:	2023-07-18
Editorial Decision:	2023-08-21
Revision Received:	2023-10-09
Editorial Decision:	2023-11-07
Revision Received:	2023-11-13
Editorial Decision:	2023-11-15
Revision Received:	2023-11-22
Accepted:	2023-11-22

Transaction Report:

August 21, 2023

Re: Life Science Alliance manuscript #LSA-2023-02281-T

Prof. Mariusz Nowacki
University of Bern
Institute of Cell Biology
Baltzerstrasse 4
Bern, BE 3012
Switzerland

Dear Dr. Nowacki,

Thank you for submitting your manuscript entitled "Structural maintenance of chromosomes (SMC) proteins are required for DNA elimination in Paramecium" to Life Science Alliance. The manuscript was assessed by expert reviewers, whose comments are appended to this letter. We invite you to submit a revised manuscript addressing the Reviewer comments.

Thank you for this interesting contribution to Life Science Alliance. We are looking forward to receiving your revised manuscript.

Sincerely,

B. MANUSCRIPT ORGANIZATION AND FORMATTING:

Reviewer #1 (Comments to the Authors (Required)):

In this interesting manuscript, Zhang and co-workers describe the role of SMC4-2 in IES excision, a critical step in Paramecium sexual reproduction.

Condensins and other SMC complexes are best known for their important role in chromosome segregation. A growing body of evidence suggests these proteins are involved in many other DNA-based processes. The findings reported here provide evidence to support the role of condensin complexes in DNA elimination.

Interestingly, the findings also suggest that this new role is unique to one of the SMC4 paralogues found in Paramecium genome, providing an interesting example of functional divergence for these complexes. Although the manuscript does not uncover the molecular details for how SMC4-2 may control IES excision, it provides interesting data on how the depletion of this gene impairs gene expression and the production of iesRNAs.

Although similar observations have been previously made in tetrahymena, confirming such non-canonical role for condensins in another ciliate species is very interesting on its own. It does not decrease the interest in this paper.

However, despite the interesting nature of these findings, the manuscript is not at a level that would meet publication standards for LSA. I would suggest several points that should be addressed.

Major issues:

1. The authors should attempt to present their work/results in a manner that is easily followed by non-experts in Paramecium biology. While the multiple types of nuclear divisions are explained in the introduction, it is hard to follow the context of these processes in the specific results. Maybe adding more schemes to the figures would facilitate. It would also help if the authors could a clear timeline for these events to understand better when some data is plotted over time (e.g. Fig. 1f).
2. The data on IES excision efficiency is very interesting and suggest a previously unknown role of a SMC4-2 in this process. However, the authors should validate their claims with two important controls:
 - a. Demonstrate that all SMCs tested are equally depleted at the various time-points of the experiments. This is critical as the claim for "functional divergence" relies on the negative result for the two SMC2s paralogues and SMC4-1. But this could be simply due to a less effective KD of these proteins. This control is even more critical for the interpretation of the double depletion results.
 - b. Demonstrate the developmental progression of the mutants. Is it conceivable that SMC4-2 mutants are simply arrested at a stage where ISE excision has not been completed. Even if unlikely, this is an alternative explanation for the obtained results that the authors should attempt to exclude. Maybe the survival data presented in Figure 1f can already address this issue. But without knowing what day 1/2/3 refer at the developmental level it is hard to judge.
3. Data in Figure 3 should be backed up by some sort of quantification. It should also be included an image of early cells where smc4-3 is absent, with a proper positive control to demonstrate microinjection efficiency.
4. The intriguing finding that co-depletion of other SMCs rescues the phenotype needs further investigation and validation. The authors draw a very general conclusions from this data ("there is an unusual relationship between SMC4-a and SMC2-1/ SMC2-2/ SMC4-1"). Unless the co-RNAi decreases the efficiency of depletion (see point #2a), these results may somehow question the authors major claim (that SMC4-2 has a direct role in IES excision). Such "unusual relationship", particularly with regards to potential protein complex partners, is often explained by competition. But without further experiments it is very difficult to interpret these results.
5. The analysis of the identified partners by Mass-spec is very interesting but in its current form it remains rather superficial. It is unclear how this data contributes an important question arising from the main results of the manuscript: is the newly uncovered role for SMC4-2 attributed to a canonical condensin complex? The authors claim that SMC4-2 co-IPs with SMC2-1. But these results do not match the observation that SMC2-1 is not defective in ISE excision. Also, it is not mentioned whether the other condensin non-SMC subunits (and potential paralogues) are enriched in the IP samples.

6. The introduction should be rewritten to better clarify what is already known prior to this work, what is a speculative idea that served as grounds for the current work and what is the summary of the present work. In the way it is written these parts are mixed and hence it is difficult to follow. For example:

- a. Lines 9-12 are highly speculative and that should be clearly stated.
- b. Lines 14-23 is unclear if this aims to summarize the current work or describe some prior findings (if the latter references are missing)
- c. Lines 25 seems a discussion of the possible importance of the findings and hence is misplaced in the introduction.

7. In the discussion, the authors refrain from comparing their data with previous observations in made in tetrahymena. I trust comparisons with previously published work should be openly discussed even if potential criticisms are politely stated. Moreover, reference to this prior work should be properly acknowledged in the introduction.

Additional comments:

1. Typo: condensin instead of condensing (several times throughout the manuscript)
2. Figure 1d legend should state the source of the data presented.
3. Panel label in Fig 3 should specify what GFP is being fused to.
4. Typo: line 29 "massive transcriptions are needed" should be "massive transcription is needed"

In summary, I think the manuscript describes a very interesting functional divergence for SMC complexes which could be of great interest in the chromosome biology and evolution fields. However, in its current form, significant controls and deeper analysis are missing to validate their claims.

Reviewer #2 (Comments to the Authors (Required)):

This manuscript by Nowacki lab describes phenotypes in developmental genome rearrangements in *Paramecium* caused by knockdown of a condensin subunit by RNAi. Condensin is a multi-subunit SMC ATPase that is crucial for mitotic chromosome condensation. It supports DNA compaction and individualization by DNA loop extrusion. *Paramecium* (and relatives) undergoes a unique programme of DNA elimination during zygotic development to generate a somatic nucleus dedicated to gene expression. The authors show that knockdown of one of two SMC4 homologs (SMC4-2) leads to a strong and genome-wide defect in DNA elimination in *Paramecium*. This implicates condensin activity in DNA elimination as previously reported (for other subunits of condensin) in *Tetrahymena*. The phenotype is likely not explained by defects in gene expression of genes needed for elimination by recombination. Curiously, the authors find that the phenotype is rescued by co-suppression of other subunits of the condensin complex. Co-IP experiments aim to reveal specific interaction partners of SMC4-2.

This work represents an interesting finding that provides further strong support for the implication of an SMC complex in the exciting biological process of DNA elimination. Several points described below however need to be carefully considered prior to publication.

Knockdown efficiency: This is particularly relevant for the double knockdowns in fig. 6 but also valid for fig. 1. How efficient is the knockdown of genes by RNAi? Is it possible that double knockdown is less efficient when compared to single knockdowns, thus providing a simple explanation for the absence of phenotypes? While it may not be trivial to test for knockdown efficiency in *Paramecium*, the authors should (at least) prominently discuss this potential caveat. An alternative explanation for the genetic rescue that should also be considered is that SMC4-2 is needed to suppress a detrimental effect of the other condensin subunits on DNA elimination.

The relation of the new findings to work performed with *Tetrahymena* (Howard-Till and Loidl, 2019) is not well described in this manuscript. The published work should be mentioned in the introduction and compared in more detail in the discussion. The authors may want to discuss whether the related functions of paralogs of condensin (if true) is the result of convergent or divergent evolution.

The authors introduce condensin in the context of chromosome condensation. However, other SMC complexes (notably cohesin) have been directly implicated in DNA recombination (for example VDJ recombination), which seems more relevant for the biological process studies here. DNA loop extrusion by cohesin is thought to regulate the selection of sites for recombination. Loop extrusion by SMC4-2 condensin may help to ensure that directly neighbouring sites (in cis) recombine for proper DNA elimination by supporting a 1D search process?

Minor comments:

The alignment in Figure 1a should include other SMC4 sequences (from closely related species and an outgroup).

Fig 2c: The labelling of the matrix seems incomplete. Top row left and right column bottom.

Fig 3a and b: Label gfp-tagged subunits directly in figure panel?

Fig 4a: A negative control for effects of the knockdown would be useful here. Same for 4b.

Fig 4b: The colour coding represents changes in gene expression but it is not explain relative to what? EV compared to what? SMC4-2 KD compared to what?

The Co-immunoprecipitation experiment presented in Fig 7 does not add much to the story in its current form. Are the expected

interaction partners found in the sample? Apart from of brief discussion of SMC2, no information is provided for other condensin subunits or known interaction partners?

The discussion is rather long and partly repeating what is also in the introduction.

Frequent typo: "condensing" instead of "condensin"

Page 5, line 13 'head-to-tail' instead of 'head-to-end'?

Page 8, line 8: It is not clear what 'as previously mentioned' means here. Remove?

Page 12, line 6: 'conserved domain architecture' instead of 'conserved domain regions'

Page 13, line 11: introduce the abbreviation 'PGM'

Re: Revised Manuscript ID LSA-2023-02281-T

On behalf of all co-authors, I would like to thank you and the two reviewers very much for further positive comments and constructive suggestions on our manuscript (MS) ID LSA-2023-02281-T. These comments and suggestions are very valuable for us to revise and improve the quality and clarity of our MS. We have revised the MS strictly according to the reviewers' comments and suggestions. We used the blue font for clarity purposes to indicate the corrections that we made in the revised manuscript. Two MS files are uploaded: with track changes, and a "clean" version. In the following section, we detail our point-by-point responses to the reviewer's comments and suggestions.

Responses to comments and suggestions of Reviewer #1:**General comments:**

In this interesting manuscript, Zhang and co-workers describe the role of SMC4-2 in IES excision, a critical step in Paramecium sexual reproduction.

Condensins and other SMC complexes are best known for their important role in chromosome segregation. A growing body of evidence suggests these proteins are involved in many other DNA-based processes. The findings reported here provide evidence to support the role of condensin complexes in DNA elimination.

Interestingly, the findings also suggest that this new role is unique to one of the SMC4 paralogues found in Paramecium genome, providing an interesting example of functional divergence for these complexes. Although the manuscript does not uncover the molecular details for how SMC4-2 may control IES excision, it provides interesting data on how the depletion of this gene impairs gene expression and the production of iesRNAs.

Although similar observations have been previously made in tetrahymena, confirming such non-canonical role for condensins in another ciliate species is very interesting on its own. It does not decrease the interest in this paper.

However, despite the interesting nature of these findings, the manuscript is not at a level that would meet publication standards for LSA. I would suggest several points that should be addressed.

Major issues:

Point 1: The authors should attempt to present their work/results in a manner that is easily followed

by non-experts in Paramecium biology. While the multiple types of nuclear divisions are explained in the introduction, it is hard to follow the context of these processes in the specific results. Maybe adding more schemes to the figures would facilitate. It would also help if the authors could a clear timeline for these events to understand better when some data is plotted over time (e.g. Fig. 1f).

Response: All figures pertaining to nuclear divisions have been appropriately explained in their respective legends. Additionally, figure legends of all tables containing chronological information have been modified.

Point 2: The data on IES excision efficiency is very interesting and suggest a previously unknown role of a SMC4-2 in this process. However, the authors should validate their claims with two important controls:

Response: We are very grateful to **Reviewer #1** for favorable comments on our MS.

Point 2a: Demonstrate that all SMCs tested are equally depleted at the various time-points of the experiments. This is critical as the claim for "functional divergence" relies on the negative result for the two SMC2s paralogues and SMC4-1. But this could be simply due to a less effective KD of these proteins. This control is even more critical for the interpretation of the double depletion results.

Responses: We thank **Reviewer #1** very much for constructive suggestions on our MS. We had to account for the influence of efficiency when performing silencing in Paramecium tetraurelia. Properly calculating the efficiency of each silencing can be challenging because individual cells within a population may experience varying levels of silencing. This variability arises because we cannot achieve a fully synchronous culture; each cell initiates autogamy at different times and reaches peak mRNA expression at different times.

To address the potential for inconsistent efficiency, we conducted two different double silencing experiments, as described in the "Single and double gene silencing of SMC2s and SMC4s" section of the MATERIALS AND METHODS, several times, and each time yielded similar results. We constructed a single plasmid containing two silencing targets to ensure consistent transcription of both gene segments. This approach maximizes the consistency of double-stranded RNA inputs.

Point 2b: Demonstrate the developmental progression of the mutants. Is it conceivable that SMC4-2 mutants are simply arrested at a stage where ISE excision has not been completed. Even if unlikely, this is an alternative explanation for the obtained results that the authors should attempt to exclude. Maybe the survival data presented in Figure 1f can already address this issue. But without knowing what day 1/2/3 refer at the developmental level it is hard to judge.

Response: All mutants went through a completely developmental progression like control. The explanation of Day1/2/3 in Figure 1f has been added. The three days counts from the end of sexual cycle which means whatever happened in autogamy, IES excision is ended, succeed or failed. It would be highlighted if a mutant stopped at a stage in the developmental rearrangement.

Point 3: Data in Figure 3 should be backed up by some sort of quantification. It should also be included an image of early cells where smc4-3 is absent, with a proper positive control to demonstrate microinjection efficiency.

Response: Since SMC4-2 is not expressed during early autogamy based on microarray and RNA-seq data, we were not able to capture any GFP signal from the microinjected cells. The same cells, when allowed to continue with the development, show GFP fluorescence appearing in the developing MACs (which is consistent with the timing of expression based on microarray and RNA-seq data. We normally do not perform a quantification of the injected construct since each individual cell receives different quantity of the plasmid. We rely on the presence/absence of GFP signal to determine whether each cell was injected successfully or not.

Point 4: The intriguing finding that co-depletion of other SMCs rescues the phenotype needs further investigation and validation. The authors draw a very general conclusions from this data ("there is an unusual relationship between SMC4-a and SMC2-1/ SMC2-2/ SMC4-1"). Unless the co-RNAi decreases the efficiency of depletion (see point #2a), these results may somehow question the authors major claim (that SMC4-2 has a direct role in IES excision). Such "unusual relationship", particularly with regards to potential protein complex partners, is often explained by competition. But without further experiments it is very difficult to interpret these results.

Response: We thank the **Reviewer #1** very much for constructive comments and suggestions. We realized this part need to be interpreted very carefully. Like clarified in **Point 2**, we tried to rule out of inconsistencies in efficiency. Repeat here again, first, we cannot calculate the efficiency; secondly, each cell may have different level of silencing and thirdly, we tried different ways and times to ensure the confidence of this co-silencing. Like Reviewer #1 said, this kind of relationship may come from competition, it is exactly the reason of performing Co-immunoprecipitation as shown in Figure 7 and this result at least supports the hypothesis the competitive relationship between SMC4-1 and SMC4-2 at specific point.

Point 5: The analysis of the identified partners by Mass-spec is very interesting but in its current form it remains rather superficial. It is unclear how this data contributes an important question arising from the main results of the manuscript: is the newly uncovered role for SMC4-2 attributed to a canonical condensin complex? The authors claim that SMC4-2 co-IPs with SMC2-1. But these results do not match the observation that SMC2-1 is not defective in ISE excision. Also, it is not mentioned whether the other condensin non-SMC subunits (and potential paralogues) are enriched in the IP samples.

Response: We are very grateful to **Reviewer #1** for favorable comments on our MS. Whether SMC4-2 contribute the canonical condensin function or not is still unclear because we did not have enough time to do further experiments. There is a hypothesis can explain the Co-IPs and co-silencing results. We assume that SMC2-1/2-2/4-1 participate in IES excision as a barrier, when the structure was intact, IES was buried into this complex; while when any part of this complex was disrupted by silencing, IESs would be released and excised. The hypothesis is not included in our MS because further experiments are needed. According to the Mass-spec results, none of non-SMC subunits are enriched which could because the Co-IPs were done with non-crosslinking samples due to technical problems. SMC4-2 Co-IP with crosslinked cells never present signals in Western blot, so we had to use non-crosslinking cells. Some interacting proteins could lose during handling.

Point 6: The introduction should be rewritten to better clarify what is already known prior to this work, what is a speculative idea that served as grounds for the current work and what is the summary

of the present work. In the way it is written these parts are mixed and hence it is difficult to follow. For example:

Response: We have clarified the current knowledge of the possible involvement of condensin in developmental genome rearrangement in ciliates (based on previous Tetrahymena studies). The data is quite limited though so it is not possible to give an extensive overview.

Point 6a: Lines 9-12 are highly speculative and that should be clearly stated.

Point 6b: Lines 14-23 is unclear if this aims to summarize the current work or describe some prior findings (if the latter references are missing)

Response: We have added the missing reference.

Point 6b: Lines 25 seems a discussion of the possible importance of the findings and hence is misplaced in the introduction.

Response: We have corrected this.

Point 7: In the discussion, the authors refrain from comparing their data with previous observations in made in tetrahymena. I trust comparisons with previously published work should be openly discussed even if potential criticisms are politely stated. Moreover, reference to this prior work should be properly acknowledged in the introduction.

Response: As mentioned above, the data on Tetrahymena condensin's involvement in genome rearrangement is very limited and quite speculative (the previous work was not specifically focusing on genome development). In addition we found some issues with control samples presented in the Tetrahymena study, which showed IES retention in the untreated control sample. This was the main reason for us to refrain from discussing the old work extensively. However, now we have modified the text to include the reference to the previous work.

Additional comments:

Point 1: Typo: condensin instead of condensing (several times throughout the manuscript)

Response: There is only one "condensing" in our MS which is a verb indicating the action of condensin. We have checked the text for possible errors.

Point 2: Figure 1d legend should state the source of the data presented.

Response: We added it in the revised MS.

Point 3: Panel label in Fig 3 should specify what GFP is being fused to.

Response: We corrected Fig. 3 accordingly.

Point 4: Typo: line 29 "massive transcriptions are needed" should be "massive transcription is needed"

Response: We have corrected this mistake and double checked our manuscript carefully.

Responses to comments and suggestions of Reviewer #2:

General comments:

This manuscript by Nowacki lab describes phenotypes in developmental genome rearrangements in *Paramecium* caused by knockdown of a condensin subunit by RNAi. Condensin is a multi-subunit SMC ATPase that is crucial for mitotic chromosome condensation. It supports DNA compaction and individualization by DNA loop extrusion. *Paramecium* (and relatives) undergoes a unique programme of DNA elimination during zygotic development to generate a somatic nucleus dedicated to gene expression. The authors show that knockdown of one of two SMC4 homologs (SMC4-2) leads to a strong and genome-wide defect in DNA elimination in *Paramecium*. This implicates condensin activity in DNA elimination as previously reported (for other subunits of condensin) in *Tetrahymena*. The phenotype is likely not explained by defects in gene expression of genes needed for elimination by recombination. Curiously, the authors find that the phenotype is rescued by co-suppression of other subunits of the condensin complex. Co-IP experiments aim to reveal specific interaction partners of SMC4-2.

This work represents an interesting finding that provides further strong support for the implication of an SMC complex in the exciting biological process of DNA elimination. Several points described below however need to be carefully considered prior to publication.

Point 1: Knockdown efficiency: This is particularly relevant for the double knockdowns in fig. 6 but also valid for fig. 1. How efficient is the knockdown of genes by RNAi? Is it possible that double knockdown is less efficient when compared to single knockdowns, thus providing a simple explanation for the absence of phenotypes? While it may not be trivial to test for knockdown efficiency in *Paramecium*, the authors should (at least) prominently discuss this potential caveat. An alternative explanation for the genetic rescue that should also be considered is that SMC4-2 is needed to suppress a detrimental effect of the other condensin subunits on DNA elimination.

Response: We needed to eliminate the influence of efficiency when discussing silencing in *Paramecium tetraurelia*. Firstly, we cannot accurately calculate the efficiency of each silencing. Secondly, individual cells may undergo silencing at different levels because we cannot achieve a fully synchronized culture. This is due to variations in the initiation of autogamy and the timing of peak mRNA expression in each cell.

To ensure consistency and rule out inconsistent efficiency, we conducted two different double silencing experiments, as described in the "Single and double gene silencing of SMC2s and SMC4s" section of the MATERIALS AND METHODS, several times, with consistent results each time. We constructed a single plasmid containing two silencing targets, which should guarantee consistent transcription of both gene segments. This approach maximizes the consistency of double-stranded RNA inputs.

We believe that Reviewer #2 shares the same idea with us regarding the function of SMC4-2 in IES excision, suggesting that SMC4-2 could compete with SMC4-1 in forming a complex with SMC2-1/SMC2-2. However, we were unable to prove this due to time constraints, so we did not include this hypothesis in our manuscript to avoid overinterpretation.

Point 2: The relation of the new findings to work performed with *Tetrahymena* (Howard-Till and Loidl, 2019) is not well described in this manuscript. The published work should be mentioned in the

introduction and compared in more detail in the discussion. The authors may want to discuss whether the related functions of paralogs of condensin (if true) is the result of convergent or divergent evolution.

Response: In the *Tetrahymena* paper, a novel condensin complex was mentioned due to the introduction of new kelisin subunits, which exhibited an IES retention phenotype under RNAi. Notably, the role of SMC4 in IES excision was not discussed in that study.

In contrast, our manuscript primarily focuses on the previously unexplored function of SMC4 in IES excision. Additionally, it's important to highlight that SMC4-2 is unique to *Paramecium tetraurelia* and *caudatum*. Prior to our research, there had been no investigation into its existence, retention phenotype, localization, or co-IPs.

Furthermore, the distinct functions of the two SMC4 proteins in IES excision had not been previously reported or discussed. As a result, we believe that a detailed comparison with the *Tetrahymena* paper is unnecessary.

Point 3: The authors introduce condensin in the context of chromosome condensation. However, other SMC complexes (notably cohesin) have been directly implicated in DNA recombination (for example VDJ recombination), which seems more relevant for the biological process studies here. DNA loop extrusion by cohesin is thought to regulate the selection of sites for recombination. Loop extrusion by SMC4-2 condensin may help to ensure that directly neighbouring sites (in cis) recombine for proper DNA elimination by supporting a 1D search process?

Response: At the outset of this project, we considered cohesin as a potential candidate for genome rearrangement in *Paramecium tetraurelia*. We conducted similar studies on cohesin subunits, including SMC1/SMC3, alongside our research on condensin. However, we found no retention phenotype when silencing SMC1/SMC3, so we did not include this information in our manuscript.

The idea that SMC4-2 could facilitate proper elimination by bringing IESs closer together is intriguing. Unfortunately, due to time constraints and the absence of relevant technologies, we were unable to conduct experiments to test this hypothesis.

Minor comments:

Point 1: The alignment in Figure 1a should include other SMC4 sequences (from closely related species and an outgroup).

Response: We could follow Reviewer #2's suggestion, but it's important to note that introducing additional SMC4 sequences might deviate from the main point of Fig. 1a. In this section, our aim is to emphasize that in *Paramecium tetraurelia*, two SMC4 proteins share nearly identical protein domains but have distinct DNA sequences, as evident in Fig. 1b. This contrast raises the possibility that SMC4-2 may serve different functions than SMC4-1 in *Paramecium tetraurelia*. Expanding the elements could shift the focus towards comparisons between *Paramecium tetraurelia* and other species.

Point 2: Fig 2c: The labelling of the matrix seems incomplete. Top row left and right column bottom.

Response: We have corrected this error and conducted a thorough review of our manuscript, addressing any other identified mistakes.

Point 3: Fig 3a and b: Label gfp-tagged subunits directly in figure panel?

Response: We have corrected the labelling.

Point 4: Fig 4a: A negative control for effects of the knockdown would be useful here. Same for 4b.

Response: We are very grateful to **Reviewer #2** for favorable comments on our MS. The EV in Fig. 4 is the negative control of the knock down. The EV represent a knock down did by same strain with SMC4-2 knockdown which containing empty vectors without any effective complementation in Paramecium genome. That is the classic negative control when doing RNAi in Paramecium study.

Point 5: Fig 4b: The colour coding represents changes in gene expression but it is not explain relative to what? EV compared to what? SMC4-2 KD compared to what?

Response: We have rewritten the figure legend like this: Figure 4. Differential expression in a SMC4-2 KD. (a) Volcano plots showing the differentially expressed genes detected in SMC4-2 KD compare to control. The most statistically significant genes are shown toward the top, with upregulated genes on right and downregulated proteins on left. Gray dots mean no significant difference, green dots represent significant only in fold change. The X-axis represents \log_2 (fold change) values and Y-axis represents $-\log_{10}$ (pval) values. (b) The top 50 most differentially expressed genes in SMC4-2 KD in comparison the EV. Light blue on the top represents two replicates of EV, the orange means two replicates of SMC4-2 KD. The red and blue colors and intensity of the boxes represent changes of gene expression based on z-transformed normalized read counts generated by DESeq2. These are the normalized counts outputted by DESeq2. And this is a comparison between SMC4-2 KD and EV normalized counts which means there is nothing relative.

Point 6: The Co-immunoprecipitation experiment presented in Fig 7 does not add much to the story in its current form. Are the expected interaction partners found in the sample? Apart from of brief discussion of SMC2, no information is provided for other condensin subunits or known interaction partners?

Response: Figure 7 serves as the theoretical foundation for the hypothesis of a competitive relationship between SMC4-1 and SMC4-2. However, due to time constraints, we were unable to conduct thorough testing, which is reflected in the limited discussion on this topic. Nevertheless, this aspect is crucial to the narrative surrounding the distinct functions of the two SMC4 proteins, primarily driven by their dynamic interactions with other subunits.

It's worth noting that our Co-immunoprecipitation experiment did not show enrichment of any non-SMC subunits. The most likely reason for this outcome is that the co-IPs were performed using non-crosslinked samples due to technical issues. We had no choice but to use non-crosslinking samples because SMC4-2 Co-IP with crosslinked cells consistently yielded unsuccessful results, potentially resulting in the loss of some interacting proteins.

Point 7: The discussion is rather long and partly repeating what is also in the introduction.

Response: We have deleted repeated part in the discussion part accordingly.

Point 8: Frequent typo: "condensing" instead of "condensin"

Response: We have checked the manuscript for possible spelling errors. At times we used “condensing” as a verb to say “to condense”.

Point 9: Page 5, line 13 'head-to-tail' instead of 'head-to-end'?

Response: We have corrected this description.

Point 10: Page 8, line 8: It is not clear what 'as previously mentioned' means here. Remove?

Response: Removed.

Point 11: Page 12, line 6: 'conserved domain architecture' instead of 'conserved domain regions'

Response: We have corrected this description.

Point 12: Page 13, line 11: introduce the abbreviation 'PGM'

Response: We have added introduction of PGM in the sentence where it is first mentioned.

We have corrected other minor mistakes and errors throughout the MS.

We have diligently addressed all the comments, and we hope that you find our manuscript now meets your expectations. We eagerly await your editorial decision.

With best wishes,

Mariusz Nowacki, PhD

Professor

Institute of Cell Biology

University of Bern

Baltzerstrasse 4

CH-3012 Bern

Switzerland

Tel: +41 31 684 46 54 (office)

+41 31 684 46 53 (lab)

Fax: +41 31 631 44 31

http://www.izb.unibe.ch/research/prof_dr_mariusz_nowacki/index_eng.html

November 7, 2023

Re: Life Science Alliance manuscript #LSA-2023-02281-TR

Prof. Mariusz Nowacki
University of Bern
Institute of Cell Biology
Baltzerstrasse 4
Bern, BE 3012
Switzerland

Dear Dr. Nowacki,

Thank you for submitting your revised manuscript entitled "Structural maintenance of chromosomes (SMC) proteins are required for DNA elimination in Paramecium" to Life Science Alliance. The manuscript has been seen by the original reviewers whose comments are appended below. While the reviewers continue to be overall positive about the work in terms of its suitability for Life Science Alliance, some important issues remain.

Our general policy is that papers are considered through only one revision cycle; however, we are open to one additional short round of revision. Please note that I will expect to make a final decision without additional reviewer input upon re-submission.

Please submit the final revision within one month, along with a letter that includes a point by point response to the remaining reviewer comments.

To upload the revised version of your manuscript, please log in to your account: <https://lsa.msubmit.net/cgi-bin/main.plex>
You will be guided to complete the submission of your revised manuscript and to fill in all necessary information.

B. MANUSCRIPT ORGANIZATION AND FORMATTING:

Sincerely,

Reviewer #1 (Comments to the Authors (Required)):

In this revised manuscript, the authors addressed some of the points raised by the referees. Most of the changes performed

relate to text changes/alterations. In that sense, the text is improved for clarity (particularly for non-experts) and their findings are better discussed in the context of prior studies. Still, prior studies by Howard-Till et al 2019 should be better acknowledged (e.g. referred to in the introduction, as suggested).

Most of the revision focused solely on changes in the text and no major changes have been accomplished at the experimental level.

Addressing the efficiency of RNAi (particularly in co-depletion) was an issue raised by both reviewers.

I still think this is an important issue as some important conclusions rely on negative data (functional divergence) or co-depletion studies (competitive relationship). The fact that all KDs result in >80% death supports an efficient KD. Yet, it is conceivable that each protein is depleted with different kinetics which would limit the interpretations reported for IES elimination. And no survival data is presented for the double depletion cases.

The authors reply to this issue with the reproducibility and the use of single plasmids for the double RNAi. None of these approaches truly address the issue. A low efficient RNAi can be low, and consistently low (a caveat to conclude negative results).

Additionally, it is well known that even if transfected with the same amount of RNAi reagent, double RNAi experiments can compromise the efficiency of depletion of both targets comparatively to single depletion (most likely due to competition for the RNAi machinery).

It is conceivable that addressing the efficiency may be difficult in this particular experimental system. And I trust such limitation should not preclude the publication of these interesting findings. But if so, the authors should properly acknowledge the limitations of their conclusions when based on negative results and co-depletion studies, as sub-optimal RNAi knock-down could confound the obtained results.

Other minor issues:

- a) Figure 3: The scale bar is missing
- b) Introduction line 91, reference to Gangi et al Science 2018 is missing

Reviewer #2 (Comments to the Authors (Required)):

The authors have chosen to respond to the reviewers' comments in a minimalistic manner. While they made useful adjustments to figure presentations and text, main concerns unfortunately remain valid:

- (1) Knock-down efficiencies may (reproducibly) vary for the different target genes, thus explaining the lack of phenotype in IES excision for condensin subunits other than SMC4-2.
- (2) Reduced knock-down efficiency of SMC4-2 when also depleting SMC2-1/2 or SMC4-1 may explain the lack of phenotype in the double depletion experiments (competition in the knock-down process rather than in condensin complex assembly). The available controls do not address this point and the issue is not acknowledged in the ms text.

This point was raised in the comments from both reviewers. The authors need to address the issues prior to publication (ideally experimentally and minimally by clearly stating these likely possibilities in the manuscript).

Re: Revised Manuscript ID LSA-2023-02281-TR

On behalf of all co-authors, I would like to thank you and the two reviewers very much for further positive comments and constructive suggestions on our manuscript (MS) ID LSA-2023-02281-TR. These comments and suggestions are very valuable for us to revise and improve the quality and clarity of our MS. We have revised the MS strictly according to the reviewers' comments and suggestions. In the following section, we detail our point-by-point responses to the reviewer's comments and suggestions.

Responses to comments and suggestions of Reviewer #1:**General comments:**

In this revised manuscript, the authors addressed some of the points raised by the referees. Most of the changes performed relate to text changes/alterations. In that sense, the text is improved for clarity (particularly for non-experts) and their findings are better discussed in the context of prior studies. Still, prior studies by Howard-Till et al 2019 should be better acknowledged (e.g. referred to in the introduction, as suggested).

Response: We thank **Reviewer #1** very much for supportive comment and constructive view on our manuscript. We referred in the introduction as suggested like this: "Specifically, a unique condensin form-condensin D, was reported to be required in the somatic nuclear maturation in *Tetrahymena thermophila* (Howard-Till et al, 2019)."

Most of the revision focused solely on changes in the text and no major changes have been accomplished at the experimental level.

Response: We were trying our best to address the critique. The first author is no longer working in the lab and we currently have no students/postdocs that would be able to take over the project to perform additional experiments.

Addressing the efficiency of RNAi (particularly in co-depletion) was an issue raised by both reviewers.

Response: We added more statements in Result and Discussion part to indicate the efficiency of RNAi could be an explanation to the difference of retention phenotype between knockdowns.

I still think this is an important issue as some important conclusions rely on negative data (functional divergence) or co-depletion studies (competitive relationship). The fact that all KDs result in >80%

death supports an efficient KD. Yet, it is conceivable that each protein is depleted with different kinetics which would limit the interpretations reported for IES elimination. And no survival data is presented for the double depletion cases.

Responses: For survival data please refer to Figure 6b. We were careful in not over-interpreting the results and discussed all possible explanations for the results.

The authors reply to this issue with the reproducibility and the use of single plasmids for the double RNAi. None of these approaches truly address the issue. A low efficient RNAi can be low, and consistently low (a caveat to conclude negative results).

Response: We added more statements to discuss another possibility of low efficient RNAi in Results and Discussion.

Additionally, it is well known that even if transfected with the same amount of RNAi reagent, double RNAi experiments can compromise the efficiency of depletion of both targets comparatively to single depletion (most likely due to competition for the RNAi machinery).

Response: We agree with the above statement. It is a general issue when performing a multiple RNAi knockdown experiment.

It is conceivable that addressing the efficiency may be difficult in this particular experimental system. And I trust such limitation should not preclude the publication of these interesting findings. But if so, the authors should properly acknowledge the limitations of their conclusions when based on negative results and co-depletion studies, as sub-optimal RNAi know-down could confound the obtained results.

Response: We have added more text to better explain the limitation of RNAi in Paramecium research.

Minor issues:

a) Figure 3: The scale bar is missing

Response: We added the scale bar.

b) Introduction line 91, reference to Gangi et al Science 2018 is missing

Response: Corrected.

Responses to comments and suggestions of Reviewer #2:

The authors have chosen to respond to the reviewers' comments in a minimalistic manner. While they made useful adjustments to figure presentations and text, main concerns unfortunately remain valid:

(1) Knock-down efficiencies may (reproducibly) vary for the different target genes, thus explaining the lack of phenotype in IES excision for condensin subunits other than SMC4-2.

Response: We added more text about the efficiency of RNAi in this research and discussed the possibility of varied efficiencies could be the reason of the phenotype.

(2) Reduced knock-down efficiency of SMC4-2 when also depleting SMC2-1/2 or SMC4-1 may explain the lack of phenotype in the double depletion experiments (competition in the knock-down process rather than in condensin complex assembly). The available controls do not address this point and the issue is not acknowledged in the ms text. This point was raised in the comments from both reviewers. The authors need to address the issues prior to publication (ideally experimentally and minimally by clearly stating these likely possibilities in the manuscript).

Response: We added more statements on this issue and discussed this possibility. We are trying to provide a new insight into the mechanism of DNA elimination, but many questions remain to be answered. Certain experiments, like for instance the double silencing, do not provide ultimate answers.

We have corrected other minor mistakes and errors throughout the MS.

We have done our best to address all comments and we sincerely hope that you find our MS revised to your satisfaction. We are looking forward to receiving your editorial decision soon.

With best wishes,

Mariusz Nowacki, PhD
Professor
Institute of Cell Biology
University of Bern
Baltzerstrasse 4
CH-3012 Bern
Switzerland

November 15, 2023

RE: Life Science Alliance Manuscript #LSA-2023-02281-TRR

Prof. Mariusz Nowacki
University of Bern
Institute of Cell Biology
Baltzerstrasse 4
Bern, BE 3012
Switzerland

Dear Dr. Nowacki,

Thank you for submitting your revised manuscript entitled "Structural maintenance of chromosomes (SMC) proteins are required for DNA elimination in Paramecium". We would be happy to publish your paper in Life Science Alliance pending final revisions necessary to meet our formatting guidelines.

- please add the Twitter handle of your host institute/organization as well as your own or/and one of the authors in our system
- please note that the titles in the system and on the manuscript file must match
- please remove Graphical Abstract from the manuscript file, upload it as a separate file, and designate it as "Graphical Abstract"
- please add an Author Contributions section to your main manuscript text
- please use the [10 author names et al.] format in your references (i.e., limit the author names to the first 10)
- please add callouts for Figure 5a to your main manuscript text
- please add your supplementary figure legend to the main manuscript text after the references section and reupload a single figure file without a title page and its legend
- the Data Availability statement mentions "Extended View and Appendix", which do not exist, so please remove. Please remove all references to an Appendix from the text.
- please include sizes next to all blots
- please add scale bar to Figure S1

A. FINAL FILES:

B. MANUSCRIPT ORGANIZATION AND FORMATTING:

Sincerely,

November 22, 2023

RE: Life Science Alliance Manuscript #LSA-2023-02281-TRRR

Prof. Mariusz Nowacki
University of Bern
Institute of Cell Biology
Baltzerstrasse 4
Bern, BE 3012
Switzerland

Dear Dr. Nowacki,

Thank you for submitting your Research Article entitled "Structural maintenance of chromosomes (SMC) proteins are required for DNA elimination in Paramecium". It is a pleasure to let you know that your manuscript is now accepted for publication in Life Science Alliance. Congratulations on this interesting work.

DISTRIBUTION OF MATERIALS:

Again, congratulations on a very nice paper. I hope you found the review process to be constructive and are pleased with how the manuscript was handled editorially. We look forward to future exciting submissions from your lab.

Sincerely,
